# TSC-mTORC1 Pathway in Postnatal V-SVZ Neurodevelopment

**DOI:** 10.3390/biom15040573

**Published:** 2025-04-12

**Authors:** David M. Feliciano, Angelique Bordey

**Affiliations:** 1Department of Biological Sciences, Clemson University, Clemson, SC 29634-0314, USA; 2Center for Human Genetics, Clemson University, Greenwood, SC 29646, USA; 3Departments of Neurosurgery, and Cellular & Molecular Physiology, Wu Tsai Institute, Yale University School of Medicine, New Haven, CT 06520-8082, USA; angelique.bordey@yale.edu

**Keywords:** neurogenesis, mTOR, TSC, mTORC1, TSC1, TSC2

## Abstract

In restricted regions of the rodent brain, neurogenesis persists throughout life, hinting that perhaps similar phenomena may exist in humans. Neural stem cells (NSCs) that reside within the ventricular-subventricular zone (V-SVZ) continually produce functional cells, including neurons that integrate into the olfactory bulb circuitry. The ability to achieve this feat is based on genetically encoded transcriptional programs that are controlled by environmentally regulated post-transcriptional signaling pathways. One such pathway that molds V-SVZ neurogenesis is the mTOR pathway. This pathway integrates nutrient sufficiency with growth factor signaling to control distinct steps of neurogenesis. Alterations in mTOR pathway signaling occur in numerous neurodevelopmental disorders. Here, we provide a narrative review for the role of the mTOR pathway in this process and discuss the use of this region to study the mTOR pathway in both health and disease.

## 1. Introduction

Multipotent neural stem cells (NSCs) generate vast quantities and types of neurons within the brain. NSCs produce most neurons embryonically; however, they persist at least until the end of infancy in discrete brain regions. Joseph Altman was the first to find groups of dividing cells in the ventricular-subventricular zone (V-SVZ), which surrounds the lateral ventricles (LVs) of the postnatal rodent brain [1]. These V-SVZ cells were labeled by the DNA precursor thymidine which was passed to cells that mature into GABAergic granule cell neurons (GCs) of the olfactory bulb (OB) [2,3,4] (Figure 1). Others identified that these V-SVZ cells behave as NSCs in rodents and generate transit amplifying cells (TACs), which are highly proliferative progenitors [5]. TACs produce neuroblasts that migrate anteriorly along the rostral migratory stream (RMS) into the OB [6]. This seminal finding pointed toward the idea that a pool of cells and factors might facilitate neurogenesis in the adult brain. While the extent to which neurogenesis persists postnatally in humans is still unclear, genetic diversity, including pathogenic variants, could prolong neurogenesis in humans. The theoretical benefit of this cellular plasticity is the molding of neuronal circuitry to adapt to environmental conditions. In theory, the tradeoff of continued cell division is the genetic vulnerability of having an increased mutational burden and susceptibility to errors in building circuits. Thus, intrinsic genetically encoded mechanisms and environmental signals must carefully balance the costs and benefits of ongoing neurogenesis. The presence of NSCs within the V-SVZ provides a unique opportunity to study signaling pathways and to model diseases. This opportunity is derived from the fact that neonatal LVs are readily accessible for the delivery of molecules, proteins, plasmids, and viruses to manipulate NSCs. Moreover, no surgery is required. Owing to their sessile nature and anatomically restricted positions, specific pools of NSCs can be manipulated to study normal physiological and pathophysiological responses. Here, we discuss the cytoarchitectonic map of the V-SVZ, the cell types generated by NSCs, and the effect of altering a critical regulator of cell and circuit development, the mTOR signaling pathway within different cell types.

## 2. Heterogeneity in the V-SVZ

The V-SVZ is a three-dimensional topographic compartment having a cytoarchitecture that is patterned and realized along a fourth dimension, time (Figure 2) [7]. Patterning of the V-SVZ is likely the remnant of the morphogenic gradients and transcriptional identities of NSCs that exist during embryogenesis [8]. The result is that different regions of the postnatal V-SVZ generate different cell types. For example, dorsal radial glia are a type of NSC in the embryonic brain that generate cortical excitatory neurons starting around embryonic day (E) 12 in mice [9]. However, in the perinatal mouse brain, they produce glia [10,11]. Later, dorsal NSCs in the V-SVZ of mice produce dopaminergic periglomerular (PG) neurons in the OB [12]. This appears directly related to precise transcriptional programs dictated by the expression of the transcription factors Pax6 and Dlx2 [12]. Dorsal NSCs may also generate some excitatory glutamatergic PGs that express the transcription factor Tbr1 in mice [13]. In comparison, mouse NSCs at the medial-septal wall produce calretinin positive PGs and GCs [14]. On the other hand, the lateral wall and dorsolateral corner of the ventricles produce OB GCs and dopaminergic PGs [14]. There are now recognized V-SVZ microdomains in mice that are responsible for generating cells having unique GC laminar positioning and morphologies [15]. For example, a small ventrally located NKX2.1 microdomain derived from the embryonic medial ganglionic eminence generates postnatal GCs [16]. The lateral ganglionic eminence (LGE) also contains microdomains that produce striatal glia and OB GCs [17]. While both populations of NSCs express Dlx5/6 in the LGE, Isl1+ cells migrate to the striatum whereas Er81+ cells generate OB GCs. Transplantation experiments in rodents further support the likelihood that the potential and capacity to generate specific cell types is eventually independent of postnatal location and mostly specified by transcriptional programs established embryonically [14]. This is analogous to the cell intrinsic mechanism of NSC specification in the cortex of mice [18]. On the other hand, the programs that regulate V-SVZ NSC pools have yet to be fully recognized. For example, PDGFRβ-positive stem cells within the septal and dorso-septal wall generate oligodendrocyte progenitor cells (OPCs) in mice [19]. In addition, a newer intraventricular localized pool of stem cells within the lumen of the lateral ventricles of mice was identified in the same study [19]. These findings indicate that diverse NSCs are found along the ventricular walls and produce different cell progeny.

## 3. Targets of Postnatal Neurogenesis

It takes nearly 1 week for neuroblasts to migrate from the rodent V-SVZ to the OB [20,21]. There, 94% of neuroblasts immediately begin to mature into GCs as evidenced by the production of basal dendrites that form synapses with centrifugal fibers and receive GABAergic input within 14 days [22,23,24]. GC apical dendrites project into the external plexiform layer, where reciprocal synapses start by 21 days, and glutamatergic input occurs approximately one month after GCs have been generated in the V-SVZ [22,23,25]. GCs are found within the GC layer, mitral/tufted cell layer, and the external plexiform layer [15]. The newly born GCs have distinct basal and apical dendrite morphologies and laminar positions in mice, albeit the developmental relationship between the morphologies of these cells has not been completely characterized [14,15]. For example, many of the morphologies overlap at different developmental periods, and shorter dendrites may reflect that these “different GCs” represent different states of the same cell, such as stages of development. However, these cells also express select proteins including calretinin and calbindin, indicating that both the morphological and molecular characteristics change.

GCs form dendrodendritic synapses within the external plexiform layer that lies beneath glomeruli (Figure 3) [26,27]. There, the dendrites of mitral cells synapse with the dendrites of GCs [28]. Most GCs are axon-less inhibitory neurons [26]. The release of glutamate from mitral/tufted cell dendrites caused by orthodromic retrograde backpropagating potential activates GCs that subsequently release GABA from their dendrites [28]. GC dendrite tiling prevents signaling between nearby mitral cells from different glomeruli. The lateral inhibition of mitral cells mediated by GC therefore increases contrast between signals. The addition of new GCs and dendrites to the circuitry optimizes olfactory processing [24]. The significance of this in mice is that continued addition of GCs in the OB is a form of structural plasticity that facilitates perceptual learning [29].

Postnatal V-SVZ NSCs may also produce additional groups of functional neurons. For example, a seminal finding found that lateral adult V-SVZ NSCs generate striatal GABAergic neurons in humans and this is reduced in Huntington’s disease [30]. The ability of V-SVZ NSCs to produce striatal neurons is evolutionarily conserved, including in rodents and rabbits. In rats, the production of GABAergic striatal neurons that are parvalbumin positive starts around postnatal (P) 9 and progressively increases for an additional 2–3 weeks whereas the production of calretinin neurons peaks at P5 [31]. Ventral V-SVZ NSCs also continually produce neurons within the nucleus accumbens of mice [32]. Striatal neurogenesis is subject to environmental regulation, including pathological insults. For example, middle cerebral artery occlusion causes neuroblasts to migrate from the SVZ into the adult striatum [33]. The mechanism by which new striatal neurons are added in the mouse may differ under normal physiological versus pathological conditions.

The human V-SVZ also appears to produce neurons until ~18 months [34,35]. An evolutionarily novel bifurcation of the human RMS called the medial migratory stream (MMS) pours into the ventromedial prefrontal cortex [34]. Details regarding the integration, maturation, and function of MMS-derived neurons are unknown. While rodents have additional postnatal waves generating GABAergic neurons that leave the RMS to populate the lower cortical layers and integrate into medial prefrontal, cingulated, and infralimbic cortices, their significance remains unclear [36]. It appears that dorsal V-SVZ NSCs in rodents can also produce cortical astrocytes [37].

## 4. Techniques for Studying V-SVZ NSCs

We will briefly summarize the use of various methods to manipulate rodent NSCs and daughter cells and their limitations. These methods include injecting components in the LVs, such as drugs or extracellular vesicles (EVs), neonatal electroporation, use of transgenic mice, and viral manipulations (Figure 4). Postnatal rodent V-SVZ NSCs have a basal fiber that projects to the vasculature and an apical projection that interdigitates between ependyma and the ventricular lumen forming the hub of a pinwheel structure [38]. This ventricular contact allows one to manipulate NSCs. Injection of different components (e.g., drugs or plasmids) into the ventricles allows for the determination of their effects on the behavior of NSCs and the production of cellular progeny. For example, the intraventricular injection of soluble factors into CSF within the LVs has been used to test the effect of growth factors on neurogenesis in rats [39]. Lipophilic dyes and fluorescent beads have also been injected into the ventricle to label NSCs and study neurogenesis in mice [40]. EVs have also been injected into the LVs of mice and were taken up by microglia. The EVs were engineered to contain specific microRNAs acting as a microglia morphogen [41]. Consistent with this, it was demonstrated that NSCs release EVs that are taken up by microglia [41]. However, there are some limitations to intraventricular injections. First, the adult LVs are surrounded by multi-ciliated ependymal cells, which may prevent efficient uptake of some components. Second, the flow of CSF throughout the ventricular system results in the diffusion of injected components, which can affect many cell types in other brain regions. An example of an affected cell type is choroid plexus epithelial cells, which generate CSF and are found within the LV. Injection of CRE recombinase fused to a TAT peptide into the lateral ventricles demonstrated the selective uptake into choroid plexus epithelial cells [42,43]. Thus, injection into the LVs has the capacity to affect cells besides NSCs. Another approach that has been extensively used is the neonatal electroporation of plasmid DNA into the ventricles of rodents [3,44,45]. While this approach lacks specificity in uptake besides being restricted to cells with ventricular contact (i.e., ependymal cells versus NCS), the use of specific promoters or the introduction of inducible plasmids into mice having CRE expression within NSCs can overcome this [46]. Another option is to sort cells or nuclei using markers to study specific cell populations but this requires generating transgenic mice and large numbers of mice due to the small size of the V-SVZ [47]. 

An important limitation of using episomal plasmids is that they are diluted in dividing cells following electroporation [48,49]. Some NSCs may retain the plasmids long-term especially if they become quiescent. Electroporation of CRE recombinase or transposases can alter genomic DNA and prevents this dilution issue [50,51]. In addition, dilution may be an advantage. For example, the electroporation of tamoxifen-inducible CRE-ER^T2^ and a conditional plasmid into NSCs is accompanied by rapid dilution from actively dividing NSCs leading to plasmids expression only in the first cohort of daughter cells, including OB GCs. Injection of tamoxifen weeks later can allow for selective recombination in mature GCs allowing one to manipulate them and distinguish their roles in the OB circuit from other cell types [52]. Another limitation of electroporation is that the injection of plasmid DNA into the LVs may cause immune reactions of intraventricular epiplexus immune cells [53]. The LVs, choroid plexus, and V-SVZ are enriched in immune cells during the perinatal period [41]. Another way to manipulate NSCs and their progeny postnatally is to use promoter-driven tamoxifen inducible CRE-ERT2 mice. A popular example is the use of mice containing the rat nestin promoter-driven CRE-ERT2 [54] since nestin is expressed in NSCs. Finally, a last approach to manipulate NSCs and neurogenesis is the use of viral targeting [40]. In mature mice, there is not extensive proliferation besides in the neurogenic zones allowing for the use of retroviruses. Replication-deficient retroviruses can be used to genomically modify NSCs [55]. This in theory will not lead to targeting quiescent NSCs, which is a limitation for studying the mechanisms that push quiescent NSCs to become active. Other types of viruses such as adenovirus were injected into the parenchyma of the V-SVZ to label NSCs and demonstrate that specific microdomains generate different types of neurons [14]. While current tools aptly allow for the labeling and manipulation of NSCs, the listed limitations necessitate the development of additional methods. For example, single cell/nuclei sequencing studies have identified several new markers, potential types, and states of NSCs. Thus, tools that use this information to more accurately manipulate NSC subpopulations would be useful. Likewise, additional tools that allow for labeling the same individual NSCs at specific times would be useful. Finally, many of the methods described here likely do not adequately label NSCs that lack contact with the LVs. For example, neonatal electroporation is unlikely to label these NSCs. Identifying ways to more efficiently manipulate these cells would be beneficial.

## 5. The mTOR Pathway

Rapamycin was originally discovered in 1965 as a compound derived from Streptomyces hygroscopicus, a bacterium found on Easter Island (also known as Rapa Nui). Initially, it was reported that rapamycin had antifungal properties and was used as an antifungal agent [56]. Later, it was shown that rapamycin works by causing the FK506-binding protein (FKBP12) to bind to and inhibit the mammalian target of rapamycin (mTOR) [57]. mTOR is a catalytic subunit of two heteromeric kinases termed mTOR complex (mTORC) 1 and 2 [58]. Rapamycin causes immediate and partial inhibition of mTORC1; however, at greater doses and over time, it may inactivate mTORC2 [57,59,60,61,62,63,64]. Each mTORC contains two mTOR kinase domains containing complex specific adapters [65,66,67]. The mTORC1 adapter is RAPTOR whereas the mTORC2 adapter is RICTOR [68,69].

mTORC1 stimulates cell growth by inducing cap-dependent mRNA translation (Figure 5). This is achieved by mTORC1 inhibitory phosphorylation of eukaryotic initiation factor 4E (eIF4E) binding protein (4EBP) [70,71] mTORC1 also phosphorylates p70S6 kinase (p70S6K), which activates the ribosomal protein S6 [60,72]. mTORC1 is activated by GTP-bound RHEB [73,74,75,76]. RHEB is inhibited by tuberin (*TSC2*) and hamartin (*TSC1*) [73,74,75,76]. Tuberin is a GTPase activating protein that causes the GTPase RHEB to hydrolyze GTP [75]. Hamartin and TBC1D7 seem to stabilize tuberin [77]. Loss of hamartin or tuberin can cause excessive mTORC1 activity and abnormal brain development as discussed below.

## 6. Tuberous Sclerosis Complex (TSC)

Patients who are afflicted with TSC are frequently born with malformations of cortical development as well as abnormal growths along the LVs that were both originally called hamartomas [78,79]. The growths along the V-SVZ are subependymal nodules (SEN), which are benign and slow-growing. Once these nodules reach a specific size, they are called subependymal giant cell astrocytomas (SEGAs) [80]. The underlying cause of TSC is inactivating mutations in the *TSC1* or *TSC2* genes found on chromosomes 9q34 and 16p13.3, respectively [81,82]. These mutations, including nonsense and missense mutations, deletions, and large rearrangements, cause the loss of *TSC1* or *TSC2* [83] and uncontrolled mTORC1 activity [83,84]. Although the events causing V-SVZ SEN and SEGA remain unclear, some mechanisms can be appreciated from studying the role of mTOR in neurogenesis.

## 7. The TSC-mTORC1 Pathway in V-SVZ Neurogenesis

During development, cellular identity is tightly linked to the expression of distinct transcription factors that promote the availability of mRNAs. Indeed, single cell and single nuclei sequencing have consistently demonstrated the importance of transcriptional programs for establishing identity. This is no different in the murine postnatal V-SVZ [85,86,87]. Since growth factors such as EGF can titrate neurogenesis, environmental signals appear to be overlayed onto genetically encoded programs [39]. Thus, post-transcriptional mechanisms, including regulation of translation, could in theory mold neurogenesis. The uncoupling of mRNA availability from translation occurs as human NSCs differentiate into neurons [88]. This is important because cells might need to rapidly stop translating NSC mRNAs. Many of the differentially translated mRNAs are regulated by mTORC1. Indeed, mTORC1 controls mRNA translation at different phases of V-SVZ neurogenesis [89]. For example, translation of the stem cell factor Sox2 mRNA is regulated by mTORC1 [89]. Persistent activation of mTORC1 causes aberrant Sox2 translation [90]. Therefore, controlling mTORC1-dependent mRNA translation occurs during neurogenesis.

The ability of extracellular factors such as EGF to regulate neurogenesis spurred investigations that identified mTORC1 activity in dividing Ki67+ cells and Mash1+ TACs produced by mouse NSCs [91]. Rapamycin treatment reduced the density of TACs and dividing cells. In culture, rapamycin also induced quiescence in neurospheres and prevented neural differentiation [91]. It is important to mention that rapamycin does not completely inhibit mTORC1 phosphorylation of all substrates [92]. Thus, how mTORC1 inhibition affects these events is unclear. Despite this, it was known that quiescent NSCs become proliferative when they express EGFR. In fact, introducing mutant active EGFR pushes quiescent NSCs (qNSCs) to divide [93]. However, tumors are not generated by mutant active EGFR expression alone [94]. Indeed, EGF infusion into the ventricle only generates V-SVZ (subependymal) polyps in rats [39]. Subsequent experiments in mice demonstrated that as NSCs go from a quiescent to an activated state, they activate mTORC1 [46]. This is consistent with the fact that tuberin is inhibited by CDK4/6 during the cell cycle and mTORC1 is activated [95]. Rapamycin also blocked NSC production of TACs and 4E-BP phosphorylation [46]. This study also demonstrated that RHEB knockdown prevents NSCs from generating TACs and neurons whereas phospho (mTOR)-resistant 4E-BP1 enhanced NSC self-renewal (Figure 6) [46]. The interpretation of these data is that RHEB activation of mTORC1 and phosphorylation of 4E-BP1 allow NSCs to undergo symmetric terminal division producing highly proliferative TACs and neurons. This occurs at the expense of self-renewing divisions and maintenance of the NSC pool. RHEB activation of mTORC1 and phosphorylation of 4E-BP1 do not cause NSCs to divide but rather to differentiate. These results indicate that mTORC1 is necessary for balancing NSC self-renewal and the production of TACs. Excessive mTORC1 activity might in theory cause NSCs to become depleted. Consistent with this hypothesis, *Tsc1* deletion from mouse V-SVZ NSCs reduces the size of the OB GC layer over time [96].

*Tsc1* deletion in qNSCs does not push NSCs to divide which is similar to findings that electroporated constitutively active RHEB [97]. Removal of *Tsc1* from V-SVZ NSCs does, however, cause widespread alterations, including V-SVZ SEN and heterotopic nodules along the RMS, in the OB, and in the striatum of mice (Figure 6) [96,98]. Mice also had ventrally located nodules and developed hydrocephalus [96]. Ventral and dorsal V-SVZ NSCs vary in mTORC1 activity [99]. Considering this, the effect of *TSC2* loss in ventral or dorsal NSCs was examined using transgenic mice expressing *Nkx2-1* or *Emx1* promoter-driven CRE. This study found that Nkx2-1 deletion of *Tsc2* preferentially caused nodules to form along the lateral V-SVZ [99]. Emx1 is expressed in NSCs that generate excitatory cortical neurons and in postnatal dorsolateral V-SVZ NSCs that generate OB GCs [100,101]. As expected, *Emx1*-CRE *Tsc1* deletion expanded the V-SVZ and RMS and caused a disorganized OB GC layer which was more spread out [102]. More *Tsc1* null dorsolateral NSCs retained BrdU or were Ki67 at early stages and this effect was reduced at later ages. This is also consistent with the effects of mTORC1 hyperactivity on balancing self-renewing cell divisions with exhausting terminal cell divisions. Likewise, loss of *Tsc2* by CRE electroporation (in conditional *Tsc2* knockout mice) also caused SENs, including those that protrude or are found within the ventricles (Figure 6) [103]. One relevant finding from this study was that striatal astrocytes that are typically generated from the V-SVZ NSCs of mice did not fully differentiate in the striatum and spuriously translated mRNAs associated with stemness following *Tsc2* deletion [103]. These immature cells appeared to aberrantly produce neurons having a striatal-like morphology. The aberrant expression of immature markers in cortical tubers is also documented in TSC patients [104]. These results further support the need to tightly regulate mTORC1 activity at precise steps of development to prevent the expression of immature proteins in differentiated cells. In addition to the loss of TSC genes, simultaneous *Tsc1*/*Pten* deletion has been performed using *Nestin*-CRE-ERT2 mice [102]. Mice injected with tamoxifen at P10 had nodular/bulbous nodules along the ventricles ~30 days later and often died likely due to cerebellar developmental complications [105]. To circumvent this issue, tamoxifen was injected later (P15 or P24) followed by examination of the brains months later. Upon inspection, SENs reminiscent of those in TSC patients had developed along the caudate nucleus of the striatum [105].

The neuroblasts that are generated from V-SVZ NSCs have low mTORC1 activity and mRNA translation in mice [46,90,91]. The presence of heterotopic clusters of neurons within the RMS following the removal of *Tsc1* from V-SVZ NSCs supports a role in migration and mirrors defects of cortical lamination caused by *Tsc1* deletion from cortical radial glia [48,98]. *Tsc1* null neuroblasts seemed to migrate slower than heterozygous neuroblasts in mice but in fact likely lost direction due to the abnormal formation of multiple processes [98]. The migratory defects were mimicked by mouse neuroblasts expressing constitutively active RHEB which also led to ectopic clusters of neurons [106]. At the molecular level, in vitro analysis of *Tsc1* null neuroblasts demonstrated that there was a reduction in autophagic flux and impaired nuclear import of the transcription factor TFEB [107]. Restoring TFEB activation in *Tsc1* null neuroblasts rescues the migratory defect [107].

Many of the neuroblasts that did not reach the OB became GCs in the RMS and their somas become enlarged following *Tsc1* or *Tsc2* removal or expression of a constitutively active RHEB using mouse neonatal electroporation [52,98,106,108]. These changes all increased mTORC1 signaling as determined by phosphorylated S6 staining. The involvement of mTOR was demonstrated by neonatal CRE electroporation of mice having conditional mTOR leading to reduction in soma sizes [109]. While removal of mTOR does not reveal which mTORC is responsible for this effect, rapamycin treatment phenocopies the reduction in size suggesting a role of mTORC1 [109].

Since neurogenesis in rodents is ongoing, it is unclear whether affecting the mTORC1 pathway in mature GCs would have the same effect as changing it in NSCs. In one study, inducible CRE-ERT2 and GFP plasmids were electroporated into neonatal V-SVZ NSCs of *Tsc2* conditional knockout mice. The plasmid DNA (indicated by GFP) was diluted from actively dividing cells and ~30 days later, mice were injected with tamoxifen leading to *Tsc2* deletion and RFP expression upon recombination in mature GCs. *Tsc2* removal had a smaller effect on mTORC1, soma size, and dendrite arbors than when recombination was initiated in neonatal V-SVZ NSCs52. This is consistent with the idea that there may be discrete periods in which TSC1/2 play more prominent roles [110]. What signals are turning TSC1/2 or mTORC1 on and off during these periods is unclear. One possibility is that amino acids such as leucine, which is transported by Slc7a5 and additional transporters, activate mTORC1 and are critical, as their loss in mice leads to OB GC death [108,111].

These studies also demonstrated that loss of *Tsc1*, *Tsc2*, or increasing RHEB activity enhanced dendrite arbors of OB GCs [52,98,112]. mTOR complex components RAPTOR and RICTOR are well known to be required for the development of mouse GC dendrite arbors [109]. mTOR inhibition by rapamycin rescues the increased growth caused by expressing constitutively active RHEB [64]. Thus, the connectivity of mutant GCs might also be changed when the TSC-mTORC1 pathway is altered. Indeed, overactive RHEB also increased the frequency of inhibitory postsynaptic currents in ectopic GCs106, suggesting an increase in GABAergic synaptic connections. While the electrophysiological properties of action potentials in these mutant OB GCs did not change, the resting membrane potential was hyperpolarized. Further experiments are needed to determine how OB neuron firing, network activity, and olfaction are affected in these models, which could yield insight into the sensory changes that occur in patients.

## 8. Therapeutic Implications for TSC Patients

The extent to which the mechanisms that regulate postnatal neurogenesis in rodents relates to that in patients, including those that have TSC, remains to be quantified. Yet, undoubtedly, there have been many important lessons learned. First is that infants certainly have ongoing V-SVZ neurogenesis. It is also clear that the loss of *TSC* genes activates mTORC1 in human NSCs [110]. Indeed, mTOR inhibitors are effective in ameliorating aspects of neurological morbidity in TSC. For example, rapamycin analogs are a primary treatment for TSC SEGAs found along the ventricles [111]. Despite their utility in TSC and a range of clinical ailments, their adaptation for persistent use is hampered by clinically relevant side effects including hyperglycemia, hyperlipidemia, and insulin resistance [112]. Side effects may also be caused by inadvertent mTORC2 inhibition caused by prolonged rapalog treatment [64]. Another limitation is that rapamycin incompletely inhibits mTORC1 phosphorylation of select substrates [90]. To overcome incomplete inhibition, mTOR ATP-competitive inhibitors were developed [113]. Although in theory, the complete inhibition of mTORC1 for a short duration might overcome the toxicity caused by prolonged mTORC2 inhibition caused by rapamycin. mTOR ATP-competitive inhibitors may indeed be well tolerated [114]. mTOR is structurally related to the lipid kinase PI3K. As might be expected, dual mTOR/PI3K inhibitors have been identified that may be useful for treating TSC [115]. However, a third generation of bisteric mTOR inhibitors holds great promise as they combine the selective targeting of mTORC1 and complete ATP pocket inhibition by tethering rapamycin to the compound MLN0128 [116]. This compound, called Rapalink-1, overcomes mutations leading to rapamycin resistance by targeting mTORC1 by both mechanisms. Rapalink-1 may be useful for the treatment of TSC SEGAs [117]. Future studies on Rapalink-1 and new mTORC1 inhibitors that have been developed for treating TSC warrant further examination.

## 9. Conclusions

In conclusion, the V-SVZ contains a mosaic of NSCs that produce different cell types owing to their embryonic ontogeny. NSCs may be easily altered, including through genetic manipulation, to study molecules that may be relevant in normal developmental processes or in specific diseases. This is exemplified by studies that have examined the role of the TSC-mTORC1 pathway in the context of V-SVZ neurogenesis.

## Figures and Tables

**Figure 1 biomolecules-15-00573-f001:**
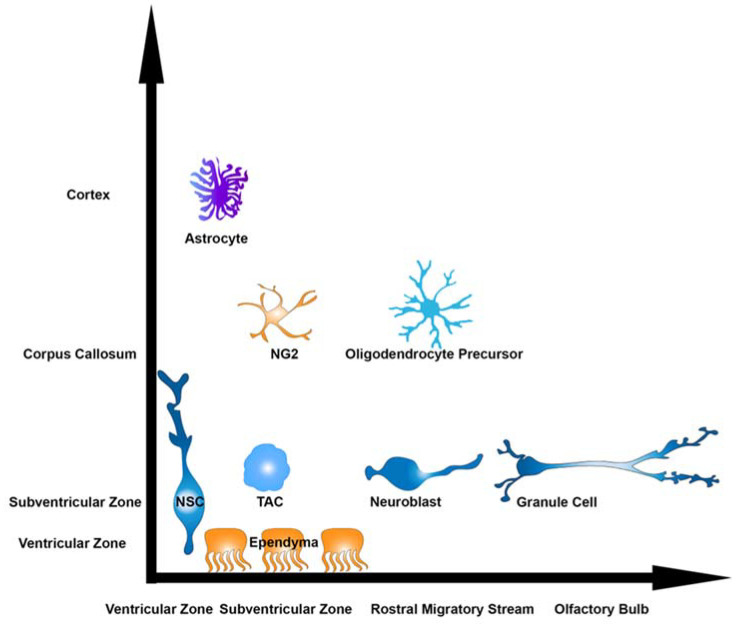
Neonatal neurogenesis. Numerous cell types are generated from the postnatal V-SVZ. NSCs may be quiescent or induced to divide and generate astrocytes, ependyma, NG2, or oligodendrocyte lineage cells. In addition, V-SVZ NSCs generate rapidly dividing transit amplifying cells (TACs), which produce neuroblasts that mature into neurons in the olfactory bulb, namely granule cells.

**Figure 2 biomolecules-15-00573-f002:**
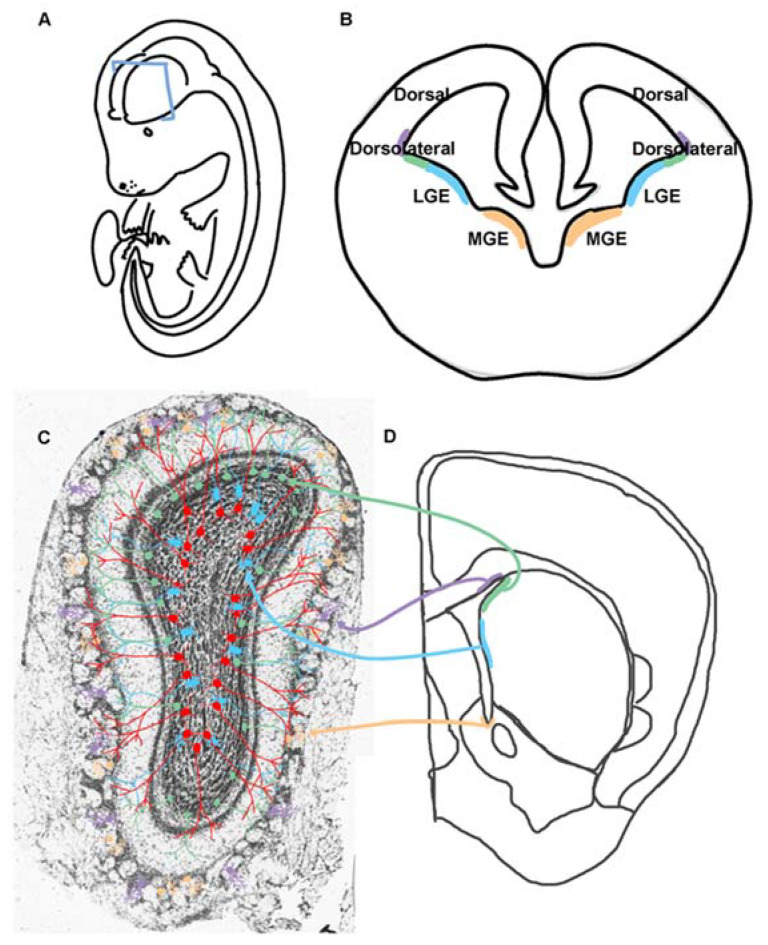
Origins of postnatal OB neurogenesis diversity. (**A**). Schematic of an embryonic mouse with a blue rectangle indicating where the coronal section of B is taken. (**B**). Coronal section of an embryonic mouse brain having V-SVZ regions labeled including the lateral and medial ganglionic eminences (LGE, MGE). The color corresponds to the respective postnatal V-SVZ regions and OB neurons that are generated. (**C**). OB neurons originating from the specific regions in the postnatal V-SVZ D and are color-coordinated with those in B. (**D**). Coronal hemi-section of an adult brain with V-SVZ regions highlighted. The identities of OB neurons can therefore be traced to specific developmental domains and transcriptional programs that dictate the identity of the NSCs found there.

**Figure 3 biomolecules-15-00573-f003:**
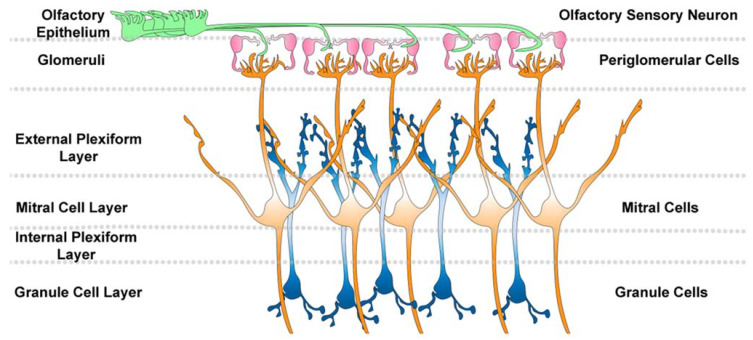
Olfactory bulb circuitry. Olfactory epithelial sensory neurons project axons onto the olfactory bulb. Thes axons form spheres called glomeruli that are surrounded by juxta/periglomerular cells. Stimulation of the sensory neurons can activate mitral cells that contain dendrites. Granule cells in the granule cell layer form dendrodendritic synapses with the mitral cells and release GABA to regulate mitral cell excitation.

**Figure 4 biomolecules-15-00573-f004:**
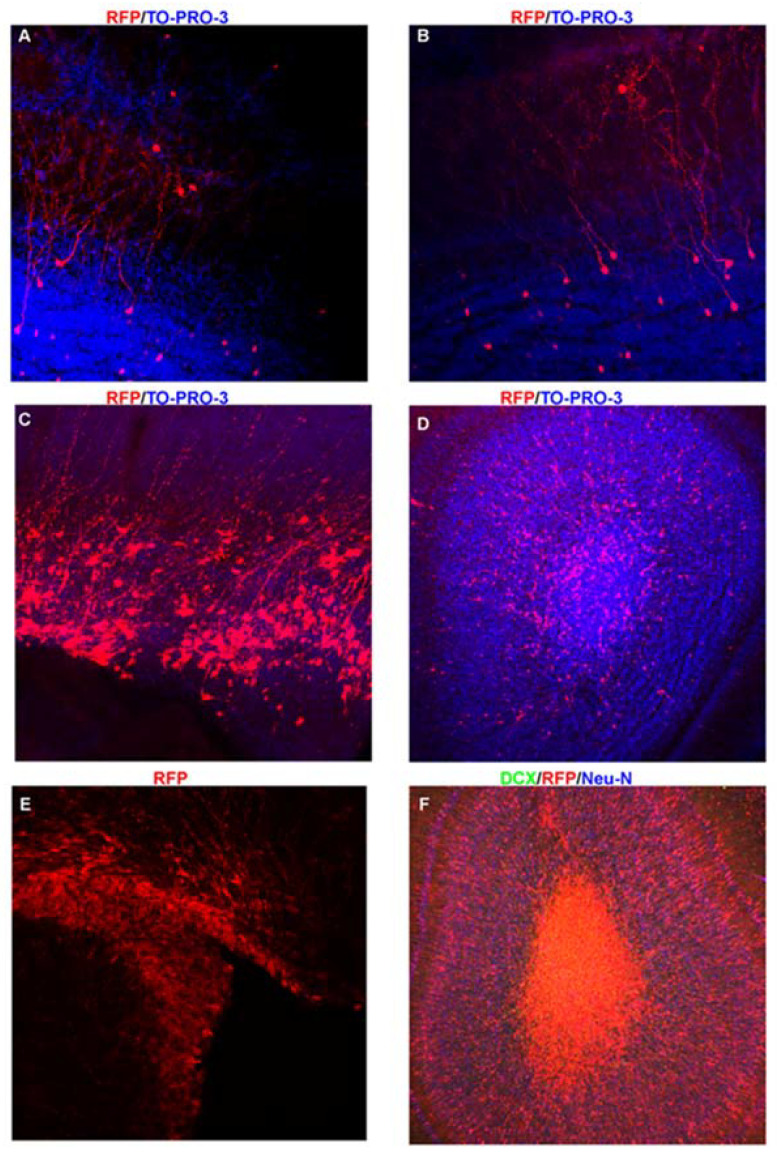
Techniques for studying V-SVZ NSCs and cell progeny. (**A**,**B**). Images of a postnatal day 30 olfactory bulb and corresponding V-SVZ cell progeny including GCs and PGs. (**C**). P6 V-SVZ dorsal region following electroporation of RFP plasmid at P0. (**D**). OB demonstrating that neuroblasts from the mouse in C are still migrating to their final destination and do not have the extensive arbors demonstrated in (**A**,**B**). (**E**). P10 V-SVZ of a *nestin*-CRE-ER^T2^ mouse crossed to an inducible RFP mouse demonstrating robust labeling of the V-SVZ. (**F**). P10 OB demonstrating that the vast majority of cells are still migrating into the OB. Note the large number of cells in the core which are DCX positive (green) and Neu-N negative (blue).

**Figure 5 biomolecules-15-00573-f005:**
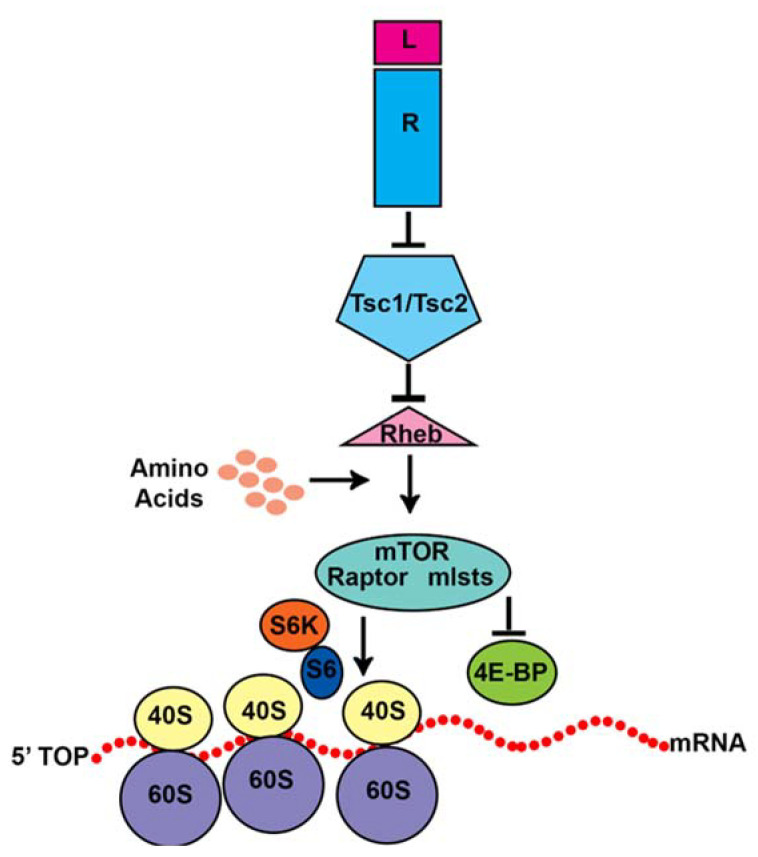
TSC-mTORC1 pathway. Ligand (L) activation of receptors (R) leads to inhibition of tuberin bound to hamartin which are encoded by *TSC2* and *TSC1*. Normally, tuberin/hamartin inhibit the GTPase RHEB activation of mTORC1. mTORC1 contains mTOR, which is a protein kinase that phosphorylates proteins to stimulate translation of mRNA containing a 5’TOP motif. In particular, mTORC1 phosphorylates and activates p70S6K. p70S6K then phosphorylates the ribosomal protein S6. mTORC1 also directly phosphorylates the eukaryotic initiation factor 4E binding protein (4E-BP), which is a translational repressor. This causes phosphorylated 4E-BP to disassemble from the eIF4F complex thus allowing translation of 5’TOP containing mRNAs.

**Figure 6 biomolecules-15-00573-f006:**
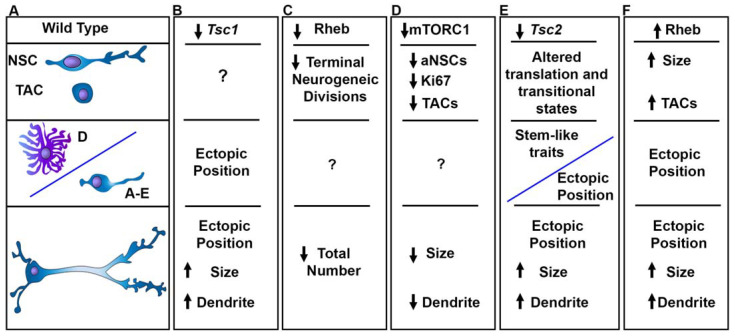
Alterations in mTORC1 signaling affect V-SVZ neurogenesis. (**A**). Wild-type NSCs go from quiescence to an activated state and produce astrocytes and/or transit amplifying cells that generate neuroblasts which mature into granule cells and periglomerular cells. (**B**). Loss of *Tsc1* leads to ectopic heterotopias and subependymal nodules as well as fewer GCs in the OB, cytomegaly, and dendrite arborization. (**C**). Reducing RHEB decreases the production of neuroblasts and decreases the total number of neurons produced. (**D**). Decreasing mTORC1 activity by rapamycin reduces the number of dividing NSCs and fewer TACs. Moreover, rapamycin treatment and RAPTOR knockdown reduce soma size and dendrite arbors. (**E**). Loss of *Tsc2* causes altered transitional states identified by single nuclei RNA sequencing and prevents proper differentiation leading to the production of partially differentiated cells expressing stem cell proteins and causes the formation of V-SVZ hamartomas and SEGA-like lesions. Moreover, OB neurons were cytomegalic and had increased dendrite growth. (**F**) Ectopic expression of mutant active RHEB increased the size of NSCs and increased the total number of TACs associated with the formation of nodules and ectopic neurons that were cytomegalic and had hypertrophic dendrites.

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
