# Peer review of "TSC-mTORC1 Pathway in Postnatal V-SVZ Neurodevelopment"

_biomolecules, 2025, doi:10.3390/biom15040573_

Round 1

Reviewer 1 Report

Comments and Suggestions for Authors

Thank you for permitting me to review this manuscript 

In this paper Neural stem cells (NSCs)  within the ventricular-subventricular zone (V-SVZ) and the mTOR pathway  in rodents have been assesed they conclude alterations in mTOR pathway signaling occur in numerous neurodevelop mental disorders.

Here are my comments , 

The authors should precise several times  through the  paper whether they are talking of human brains or rodents since this can confuse the reader 

Should  this be  a narrative review , then it should be stated 

Figure 2 is not well explained I think line 95 should appear around line91

Please list some perspectives in future human studies 

Author Response

POINT BY POINT RESPONSE

Introduction. We thank the reviewers for providing feedback for our review. Responses are provided in red under individual comments. We thank the reviewers for their general agreement that “the manuscript is well-written” (Reviewer 2) and that the review is “highly relevant” (Reviewer 3). There was consensus that additional information regarding “perspectives in future human studies” (Reviewer 1), “future therapeutic approaches in the field” (Reviewer 2), and that the “discussion of therapeutic implications could be expanded” (Reviewer 3). To address these comments, we now provide a section called “Therapeutic Implications for TSC Patients”.

Respectfully,

Dave and Angélique

Reviewer 1.

In this paper Neural stem cells (NSCs)  within the ventricular-subventricular zone (V-SVZ) and the mTOR pathway  in rodents have been assesed they conclude alterations in mTOR pathway signaling occur in numerous neurodevelop mental disorders.

Here are my comments, The authors should precise several times  through the  paper whether they are talking of human brains or rodents since this can confuse the reader 

Response. We have added the requested information where relevant.

Should this be  a narrative review, then it should be stated 

Response. We have added this to the abstract.

Figure 2 is not well explained I think line 95 should appear around line91

Response. We have moved line 95 accordingly and expanded the explanation.

Please list some perspectives in future human studies 

Response. A complementary section is added titled, “Therapeutic Implications for TSC Patients”.

Reviewer 2 Report

Comments and Suggestions for Authors

I have read the manuscript entitled "TSC-mTORC1 Pathway in Postnatal V-SVZ Neurodevelopment" by David M. Feliciano and Angelique Bordey which analyzes the process of neurogenesis in the ventricular-subventricular zone (V-SVZ) of the brain, which persists throughout life in rodents and possibly humans. It seems that mTOR pathway is a key factor in the above process. The authors present different methodologies for the manipulation of NSCs that make the evaluation of neurodevelopmental processes possible. Finally, the article highlights the importance of genes such as TSC1, TSC2 in regulating the mTORC1 activity during neurogenesis.

Generally, the manuscript is well-written and well-structured with good overall reading flow among sections. The manuscript provides sufficient evidence for the role of mTOR and the TSC-mTORC1 pathways. Furthermore, there is a detailed description of various methodologies like intraventricular injections and neonatal electroporation but I believe that their limitations and potential future directions should be presented as well. Finally, in the section where the implications of mTOR in neurodevelopmental disorders, the authors could include some content related to promising or future therapeutic approaches in the field.

To my opinion, this manuscript seems to be a detailed and well-written work associated with neurogenesis.

Author Response

POINT BY POINT RESPONSE

Introduction. We thank the reviewers for providing feedback for our review. Responses are provided in red under individual comments. We thank the reviewers for their general agreement that “the manuscript is well-written” (Reviewer 2) and that the review is “highly relevant” (Reviewer 3). There was consensus that additional information regarding “perspectives in future human studies” (Reviewer 1), “future therapeutic approaches in the field” (Reviewer 2), and that the “discussion of therapeutic implications could be expanded” (Reviewer 3). To address these comments, we now provide a section called “Therapeutic Implications for TSC Patients”.

Respectfully,

Dave and Angélique

I have read the manuscript entitled "TSC-mTORC1 Pathway in Postnatal V-SVZ Neurodevelopment" by David M. Feliciano and Angelique Bordey which analyzes the process of neurogenesis in the ventricular-subventricular zone (V-SVZ) of the brain, which persists throughout life in rodents and possibly humans. It seems that mTOR pathway is a key factor in the above process. The authors present different methodologies for the manipulation of NSCs that make the evaluation of neurodevelopmental processes possible. Finally, the article highlights the importance of genes such as TSC1, TSC2 in regulating the mTORC1 activity during neurogenesis.

Generally, the manuscript is well-written and well-structured with good overall reading flow among sections.

Response. We appreciate your comment.

The manuscript provides sufficient evidence for the role of mTOR and the TSC-mTORC1 pathways. Furthermore, there is a detailed description of various methodologies like intraventricular injections and neonatal electroporation but I believe that their limitations and potential future directions should be presented as well. 

Response related to limitations. We agree with the reviewer’s comment, but we are a little uncertain about which limitations to add because we have already listed extensive limitations as described below. We would appreciate it if the reviewers had any other limitations in mind that we should add. Nevertheless, the next section related to “potential future directions” indirectly highlight limitations of the present tools. 

 “However, there are some limitations to intraventricular injections. First, the adult LVs are surrounded by multi-ciliated ependymal cells, which may prevent efficient uptake of some components.”

“Second, the flow of CSF throughout the ventricular system results in the diffusion of injected components, which can affect many cell types in other brain regions. An example of an affected cell type are choroid plexus epithelial cells, which generate CSF and are found within the LV. Injection of CRE recombinase fused to a TAT peptide into the lateral ventricles demonstrated the selective uptake into choroid plexus epithelial cells. Thus, injection into the LVs has the capacity to affect cells besides NSCs.”…

”… While this approach lacks specificity in uptake besides being restricted to cells with ventricular contact…”

“…but this requires generating transgenic mice and large numbers of mice due to the small size of the V-SVZ. An important limitation of using episomal plasmids is that they are diluted in dividing cells following electroporation.”

“…Another limitation of electroporation is that the injection of plasmid DNA into the LVs may cause immune reactions of intraventricular epiplexus immune cells53.”

“This in theory will not lead to targeting quiescent NSCs, which is a limitation for studying the mechanisms that push quiescent NSCs to become active.”

…and potential future directions should be presented as well. 

Response. We have added the following future directions to this section: “While current tools aptly allow for the labeling and manipulation of NSCs, the listed limitations necessitate the development of additional methods. For example, single cell/nuclei sequencing studies have identified several new markers, and types and states of NSCs. Thus, tools that use this information to more accurately manipulate NSC subpopulations would be useful. Likewise, additional tools that allow for labeling the same individual NSCs at specific times would be useful. Finally, many of the methods described here likely do not adequately label NSCs that lack contact with the LVs. For example, neonatal electroporation is unlikely to label these NSCs. Identifying ways to more efficiently manipulate these cells would be beneficial.”

Finally, in the section where the implications of mTOR in neurodevelopmental disorders, the authors could include some content related to promising or future therapeutic approaches in the field.

Response. We now include the requested information under a new section titled, “Therapeutic Implications for TSC Patients”.

To my opinion, this manuscript seems to be a detailed and well-written work associated with neurogenesis.

Response. Thank you for your comment.

Reviewer 3 Report

Comments and Suggestions for Authors

This review provides a comprehensive and well-structured analysis of the role of the TSC-mTORC1 pathway in postnatal ventricular-subventricular zone (V-SVZ) neurodevelopment. The manuscript successfully integrates molecular, cellular, and functional insights while highlighting pathophysiological implications in neurodevelopmental disorders. Given the increasing interest in mTOR signaling and neurogenesis, this review is highly relevant for researchers in neuroscience, developmental biology, and molecular medicine.

Areas for Improvement

Clarity in Mechanistic Pathways - The manuscript contains dense descriptions of mTOR signaling and neurogenesis, which may be difficult for non-specialist readers.
Consider adding a summary figure or table outlining key molecular interactions in the TSC-mTORC1 pathway.

Discussion of Therapeutic Implications Could Be Expanded - While the review covers mTOR inhibitors (e.g., rapamycin), the discussion on translational applications and clinical trials is relatively brief.
Are there ongoing clinical trials targeting mTORC1 in neurodevelopmental disorders?
What are the challenges of using mTOR inhibitors in pediatric or adult patients?
A table summarizing potential mTOR-targeted therapies and their preclinical/clinical status would improve clarity.

Formatting and Readability Enhancements - Some paragraphs contain long, complex sentences.
Breaking down key concepts into shorter, more digestible subsections would improve readability.

Comments on the Quality of English Language

The English could be improved to more clearly express the research.

Author Response

POINT BY POINT RESPONSE

Introduction. We thank the reviewers for providing feedback for our review. Responses are provided in red under individual comments. We thank the reviewers for their general agreement that “the manuscript is well-written” (Reviewer 2) and that the review is “highly relevant” (Reviewer 3). There was consensus that additional information regarding “perspectives in future human studies” (Reviewer 1), “future therapeutic approaches in the field” (Reviewer 2), and that the “discussion of therapeutic implications could be expanded” (Reviewer 3). To address these comments, we now provide a section called “Therapeutic Implications for TSC Patients”.

Respectfully,

Dave and Angélique

Reviewer 3.

This review provides a comprehensive and well-structured analysis of the role of the TSC-mTORC1 pathway in postnatal ventricular-subventricular zone (V-SVZ) neurodevelopment. The manuscript successfully integrates molecular, cellular, and functional insights while highlighting pathophysiological implications in neurodevelopmental disorders. Given the increasing interest in mTOR signaling and neurogenesis, this review is highly relevant for researchers in neuroscience, developmental biology, and molecular medicine.

Response. Thank you for your comment.

Areas for Improvement

Clarity in Mechanistic Pathways - The manuscript contains dense descriptions of mTOR signaling and neurogenesis, which may be difficult for non-specialist readers.

Consider adding a summary figure or table outlining key molecular interactions in the TSC-mTORC1 pathway.

Response. We apologize for the complex description. We have simplified this section and better highlighted Figure 5, which summarizes this pathway.

Discussion of Therapeutic Implications Could Be Expanded - While the review covers mTOR inhibitors (e.g., rapamycin), the discussion on translational applications and clinical trials is relatively brief.

Response. Since this manuscript is a part of a special issue focused on Neurodevelopment, we intended to avoid excessive discussion of the important translational and clinical applications. However, we have added a new section “Therapeutic Implications for TSC Patients” that describes the relevance of the TSC-mTOR pathway in neurogenesis as it relates to TSC patients.

Are there ongoing clinical trials targeting mTORC1 in neurodevelopmental disorders?
What are the challenges of using mTOR inhibitors in pediatric or adult patients?

Response. We have added commentary regarding mTOR inhibitors and the challenges of their use as it relates to TSC under a new section, “Therapeutic Implications for TSC Patients”.

A table summarizing potential mTOR-targeted therapies and their preclinical/clinical status would improve clarity.

Response. Interestingly, while there are clinical trials for mTOR targeted therapies, most are for rapalogs and are more pertinent to the cerebral cortex. We have added that rapalogs are useful for TSC SEGAs and briefly discuss some of the limitations and opportunities and highlight some preclinical findings of additional drugs.

Formatting and Readability Enhancements - Some paragraphs contain long, complex sentences.
Breaking down key concepts into shorter, more digestible subsections would improve readability.

Response. We have worked with the team to further break down sentences.